# Modeling Unveils How Kleptoplastidy Affects Mixotrophy Boosting Algal Blooms

**DOI:** 10.3390/biology14070900

**Published:** 2025-07-21

**Authors:** Irena V. Telesh, Gregory J. Rodin, Hendrik Schubert, Sergei O. Skarlato

**Affiliations:** 1Zoological Institute of the Russian Academy of Sciences, 199034 St. Petersburg, Russia; 2Department of Aerospace Engineering and Engineering Mechanics, The University of Texas at Austin, Austin, TX 78712, USA; gjr@oden.utexas.edu; 3Institute for Biosciences, Faculty of Mathematics and Natural Sciences, The University of Rostock, 18051 Rostock, Germany; 4Institute of Cytology of the Russian Academy of Sciences, 194064 St. Petersburg, Russia; sergei.skarlato@mail.ru

**Keywords:** dinoflagellates, ecological indicator, harmful algae bloom, kleptoplastidy, mixotrophy, model, *Prorocentrum cordatum*, protists

## Abstract

While it is well known that planktonic algae often form blooms that cause harm to ecosystems globally due to oxygen depletion and/or intoxication, there is a lot to be learned about the underlying mechanisms, such as mixotrophy. Mixotrophic bloom-forming algae are highly adaptable to environmental fluctuations due to flexibility of feeding strategies. Kleptoplastidy is an essential component of mixotrophy. This phenomenon implies that cells of mixotrophic protists acquire, maintain, and exploit chloroplasts of prey microalgae. We present a new mathematical model, which for the first time demonstrates how kleptoplastidy may accelerate reproductive growth of mixotrophic protists and thus lead to algal blooms. A key ingredient of this model is a new ecological indicator, a kleptoplastidy index, introduced as a measure of contribution of kleptoplastidy to mixotrophy. The model was applied to potentially toxic bloom-forming mixotrophic dinoflagellates. Its predictions support the hypothesis that kleptoplastidy can significantly increase the division rate of algae. Thus, kleptoplastidy in mixotrophic protists boosts their population growth and leads to biomass accumulation and blooms. The proposed model may become an essential building block in mathematical modeling of basic ecological processes of utmost importance. It will contribute to forecasting harmful algae blooms that deteriorate marine coastal environments worldwide.

## 1. Introduction

Kleptoplastidy is one of the most intriguing though poorly studied aspects of mixotrophic feeding mode, which theoretically can boost reproduction of planktonic algae capable of harmful blooms. As a composite nutrition strategy, mixotrophy is exploited by many protists, including globally distributed dinoflagellates, some of which are toxic or potentially toxic and form harmful algae blooms (HABs), representing a serious threat to sea coastal environments worldwide [1,2,3,4].

Mixotrophy is a combination of autotrophy and heterotrophy. The former is based on photosynthesis and dissolved inorganic nutrients. The latter is realized in two forms. First, osmotrophy, or consumption of dissolved organic nutrients by osmosis. Second, phagotrophy, or consumption of prey cells of algae and/or bacteria by engulfing them. At present, photo-osmo-mixotrophy is viewed as a ubiquitous feeding strategy [4], whereas photo-phago-mixotrophy is less common and its realization varies significantly across the protist spectrum [5]. Once combined, the two strategies give rise to remarkable adaptability of mixotrophic protists to fluctuating environments.

Plankton capable of mixotrophic feeding is referred to as mixoplankton [6]. It is characterized by high environmental tolerance [7], as it can advantageously exploit broad ecological niches including diverse food spectra [8,9,10]. As a result, mixoplankton is often responsible for HABs in aquatic ecosystems [11,12,13,14]. Due to mixoplankton toxicity, or potential toxicity, HABs pose a major global threat, as they negatively affect aquatic communities and ecosystems, fisheries, aquaculture and wildlife, coastal tourism, and human health [15,16,17]. Therefore, the knowledge of kleptoplastidy role in boosting mixotrophy and thus promoting algal blooms is crucial. Meanwhile, for the majority of bloom-forming species, such information is very scarce or lacking.

This paper is concerned with kleptoplastidy (or kleptoplasty) of the protistan mixoplankton, a feeding mode in which a mixotrophic protist cell acquires, maintains, and exploits chloroplasts of prey cells of other algae as photosynthesis reactors [18,19]. Here, we hypothesize that kleptoplastidy can significantly increase the division rate of mixotrophic bloom-forming protists.

Following our perspective on the characterization of cellular and molecular drivers and environmental triggers leading to HABs [10,20,21,22,23], in this paper, we develop a mathematical (as opposed to simulation-based) model of kleptoplastidy. We rely on basic cellular mechanisms for developing the model and use this model for quantifying the contribution of kleptoplastidy to reproductive growth of mixotrophic protists. Finally, we focus on the development of a new ecological indicator, a kleptoplastidy index, as a function of basic eco-physiological variables (cells division rate and consumption rates of different resources). To the best of our knowledge, the proposed model has no counterparts in the pertinent literature, and neither such indicators nor models and experimental data suitable for model calibration and validation are widely available so far [24,25,26,27,28,29,30]. We believe that, as a broader implication, our model may be incorporated in the future ecosystems’ assessment strategies, and the proposed approach can significantly improve existing mixotrophy models, HABs forecasting, and environmental management.

## 2. Materials and Methods

The mathematical structure of the model is based on extending basic Monod’s equation [31] to a biological system consisting of a single protist, one inorganic resource, and several identical prey cells containing chloroplasts. This approach results in governing equations with two advantageous properties. First, those equations are sufficiently simple to be solved analytically. Second, the equations account for kleptoplastidy via measurable parameters representing essential morpho-physiological features of a mixotrophic protist: cells’ division rate and consumption rates of different resources. A kleptoplastidy model was developed by introducing the concept of biological time and three thought experiments involving controlled environments, which can be realized in laboratory settings only. The first experiment was designed so that only the autotrophic mode was active; in the second, only heterotrophic mode was active; and in the third experiment, all three modes, the autotrophic, heterotrophic, and kleptoplastidic, were active. Consequently, in processing experimental results, it was possible to isolate the kleptoplastidic feeding mode and evaluate its contribution to cells’ growth.

For demonstration purposes, the model is applied to the dinoflagellate *Prorocentrum cordatum* (Ostenfeld) J.D. Dodge, 1975. This mixotrophic protist has been investigated by several research teams [3,32,33,34,35,36] as it is potentially toxic and globally spread. Recently, *P. cordatum* was singled out as the most common, globally distributed mixotrophic protist responsible for HABs in 30 countries [3]. A suitable organic resource for studying kleptoplastidy in *P. cordatum* is the cryptophyte microalga *Teleaulax* sp. [3] as this organism contains chloroplasts.

Our research aims at the development of a kleptoplastidy model by combining generality and simplicity of models in physics with specificity of biological descriptions. For this reason, the model developed in Section 3.1 and Section 3.2 is for a generic mixotrophic protist. In contrast, in Section 3.3, the model is applied specifically to *Prorocentrum cordatum* and *Teleaulax* sp. The basic symbols used in the model and their explanations and units are given in Table 1. Details that are more specific along with the model description are provided in Section 3.1, Section 3.2 and Section 3.3.

## 3. Results

### 3.1. Model

#### 3.1.1. Background of the Model

The objective of this section is to develop a basic model of kleptoplastidy for a conventional reference mixotrophic protist. To this end, the protist is viewed simply as a vessel containing native and acquired chloroplasts. Native chloroplasts form a discontinuous belt adjacent to the cell wall interior and function as photosynthesis reactors, using light and inorganic resources. Acquired chloroplasts function as additional photosynthesis reactors, and we refer to them as kleptoplasts. In addition, we identify a free volume available for accumulating kleptoplasts (Figure 1).

A prey cell is viewed as a source of easily digestible organic nutrients and laboriously digestible chloroplasts. Once a prey cell is captured, easily digestible components are rapidly metabolized in food vacuoles, while laboriously digestible chloroplasts become kleptoplasts. This cycle is repeated for subsequently captured prey cells. The ensuing accumulation of kleptoplasts and their use as additional photosynthesis reactors for reproductive growth of the protist is the essence of kleptoplastidy and the subject of our model.

The entire modeling setup is presented in Figure 1. There, the protist is shown as the large irregular centrally located shape, inorganic resources as small peripheral gray circles, and prey cells as large peripheral circles. Native chloroplasts of the protist are shown in blue, while native chloroplasts of prey cells are shown in red, and kleptoplasts are shown in red with dark outline, which represents the additional membrane of the former food vacuole. The free volume available for accumulating kleptoplasts is shown in green (Figure 1).

#### 3.1.2. Monod’s Equation and the Maximum Cell Division Rate

The most fundamental model of reproductive growth of microorganisms is based on Monod’s equation [31]. In particular, if the system of interest consists of organisms (cells) of a single species and a single resource, the cells divide at the rate(1)$$\mu =\Omega \frac{R}{K+R}$$

Here, Ω and K are system-specific positive parameters, and R is the environmental resource concentration. This equation implies(2)$$\mu <\mu_{\max}=\Omega$$
and therefore Ω is referred to as the maximum cell’s division rate. The parameter K is referred to as the half-saturation constant.

For our purposes, it is useful to introduce a normalized concentration, defined as(3)$$r:=\frac{R}{K},$$
so that (1) can be rewritten as(4)$$\mu =\Omega \frac{r}{1+r}.$$

We define resource-rich environments in which the normalized resource concentration *r* >> 1. Then, (4) implies that resource-rich environments are necessary for realizing division rates close to $\mu_{\max}=\Omega$.

For a single species and two substitutable resources [37], (4) is generalized as(5)$$\mu =\Omega_{1}\frac{r_{1}}{1+r_{1}}+\Omega_{2}\frac{r_{2}}{1+r_{2}}.$$

Here, $r_{1}$($r_{2}$) is the normalized concentration, defined in (3), and $\Omega_{1}$($\Omega_{2}$) is the maximum cell’s division rate due to the first (second) resource. According to (5), the maximum cell’s division rate is(6)$$\mu_{\max}=\Omega_{1}+\Omega_{2}.$$

In this equation, $\mu_{\max}$ is associated with the optimal way of consuming the two resources.

In the context of mixotrophy, it is meaningful to regard the resources in (5) and (6) as inorganic and organic. Accordingly, we assign the subscripts 1 and 2 to the inorganic and organic resources, respectively. In what follows, it will be demonstrated that (6) underestimates the true maximum division rate for mixotrophic protists capable of kleptoplastidy. To this end, we will develop a kleptoplastidy model by introducing the concept of biological time and three theoretical thought experiments.

#### 3.1.3. Biological Time

Biological time is a useful concept that allows one to analyze scenarios in which the cells’ division rate is time-dependent, as it turns out to be the case in the proposed kleptoplastidy model. It is defined as a dimensionless function of time, $\psi \left(t\right)$, which satisfies the differential equation(7)$$\frac{d\psi }{dt}=\mu .$$

Further, biological time is confined to the interval 0≤ψ≤1, with ψ=0 and ψ=1 corresponding to the beginning and end of a biological process of interest, respectively. For our purposes, the process of interest occurs between two consecutive divisions of the reference protist cell. Mathematical foundations of this approach have been firmly established in the literature concerned with predicting service life of structural components [38].

As a demonstration, let us consider the most basic situation, where μ=Ω is constant:(8)$$\frac{d\psi }{dt}=\Omega .$$

Integration of this equation with respect to time yields(9)$$\int_{0}^{1}d\psi =\int_{0}^{T}\Omega dt.$$

Here, T is the time interval between two consecutive divisions. This equation yields(10)1=ΩT
and(11)$$\;T=\frac{1}{\Omega }.$$

Of course, the answer for T could have been obtained immediately, without introducing biological time. But this is because, in this example, the right-hand side of (7) is independent of time, t. In general, one needs integration, and this will be the case with analysis of kleptoplastidy.

#### 3.1.4. Thought Experiments

In this subsection, we present three thought experiments leading to the kleptoplastidy model. The first two experiments provide essential building blocks, while the third experiment reveals a synergistic contribution of kleptoplastidy.

##### First Experiment: Autotrophy

Let us consider a single reference mixotrophic protist in an environment characterized by the normalized resource concentrations *r*_1_ >> 1 and $r_{2}=0$. That is, the environment is inorganic-resource-rich but it contains no organic resources. Suppose that the outcome of this experiment is the division rate $\mu^{\left(1\right)}$. Here, we use the superscript (1) to refer to the first thought experiment. For this experiment, (5) implies(12)$$\Omega_{1}=\mu^{\left(1\right)},$$
so that the measurement $\mu^{\left(1\right)}$ yields the parameter $\Omega_{1}$. Note that, in the chosen environment, the protist behaves like an autotroph.

For future reference, let us define the maximum division rate due to a unit native chloroplast mass as(13)$$\omega_{1}:=\frac{\Omega_{1}}{M_{1}}.$$

Here $M_{1}$ is the mass of native chloroplasts in the protist. We use the term maximum, as $\omega_{1}$ is directly related to the maximum division rate $\Omega_{1}$.

##### Second Experiment: Heterotrophy

Let us consider the protist in an environment characterized by $r_{1}=0$ and *r*_2_ >> 1. That is, the environment contains no inorganic resources, but it is organic-resource-rich. Further, the organic resources are prey cells containing chloroplasts. Suppose that the outcome of this experiment is the division rate $\mu^{\left(2\right)}$. Here, we use the superscript (2) to refer to the second thought experiment. For this experiment, (5) implies(14)$$\Omega_{2}=\mu^{\left(2\right)},$$
so that the measurement $\mu^{\left(2\right)}$ yields the parameter $\Omega_{2}$. Note that, in the chosen environment, the protist behaves like a heterotroph.

Since there are no inorganic resources, kleptoplasts do not contribute to reproductive growth. Furthermore, kleptoplasts may halt reproductive growth by fully occupying the free space (Figure 1), so that the protist can no longer consume prey cells. We assume that this does not happen, and the protist steadily consumes prey cells.

Let us define τ as the time it takes for the protist to digest one prey cell. During this time, two events occur. First, the protist acquires essentially indigestible native chloroplasts of the prey cell, and they become its kleptoplasts. Second, the protist metabolizes easily digestible remaining nutrients of the prey cell.

In the adopted scenario, acquisition of kleptoplasts is a cumulative process, whose rate is(15)$$\frac{d{\hat{M}}_{1}}{dt}=\frac{m_{1}}{\tau }.$$

Here, ${\hat{M}}_{1}$ is the mass of kleptoplasts and $m_{1}$ is the mass of native chloroplasts in one prey cell. By assuming that the initial mass of kleptoplasts is zero, integration of (15) yields(16)$${\hat{M}}_{1}=\frac{m_{1}t}{\tau }.$$

##### Third Experiment: Mixotrophy

Let us consider the protist in an environment characterized by the normalized resource concentrations *r*_1_ >> 1 and *r*_2_ >> 1. That is, the environment is rich with both inorganic and organic resources. As in the second thought experiment, the organic resource are prey cells containing chloroplasts. Suppose that the outcome of this experiment is the division rate $\mu^{\left(3\right)}$. Here, we use the superscript (3) to refer to the third thought experiment. For this environment, (5) implies$$\mu^{\left(3\right)}=\Omega_{1}+\Omega_{2},$$
but this would be incorrect, as (5) does not account for kleptoplastidy. Rather, we write(17)$$\mu^{\left(3\right)}=\Omega_{1}+\Omega_{2}+\hat{\mu },$$
where $\hat{\mu }$ is the division rate representing photosynthetic activities of kleptoplasts, which were present but inactive in the second thought experiment.

Using results of the first two thought experiments, we can combine (13) and (16) to estimate(18)$$\hat{\mu }=\omega_{1}^{*}{\hat{M}}_{1}=\omega_{1}^{*}\frac{m_{1}t}{\tau }=\gamma \omega_{1}\frac{m_{1}t}{\tau }=\frac{\gamma \Omega_{1}m_{1}t}{M_{1}\tau }.$$

Here, $\omega_{1}^{*}$ is the maximum division rate due to a unit kleptoplasts mass, and $\gamma :=\omega_{1}^{*}/\omega_{1}$. While (18) is useful as it relates kleptoplastidy to morpho-physiological parameters, it may be prohibitively demanding on a required experimental program. Therefore, it is prudent to replace (18) with(19)$$\hat{\mu }=\beta t,$$
and treat β as a phenomenological parameter. This parameter can be determined from the measurements using (12), (14), (17), and (19):(20)$$\beta =\frac{\mu^{\left(3\right)}-\mu^{\left(1\right)}-\mu^{\left(2\right)}}{t}.$$

Of course, if desired, β can be related to the morpho-physiological parameters in (18):(21)$$\beta =\frac{\gamma \Omega_{1}m_{1}}{M_{1}\tau }.$$

Equations (7), (17) and (19) imply that biological time in the third thought experiment is(22)$$\frac{d\psi }{dt}=\mu^{\left(3\right)}=\Omega_{1}+\Omega_{2}+\beta t.$$

Upon integration of this equation, as it was carried out in (9), we obtain(23)$$1=\left(\Omega_{1}+\Omega_{2}\right)T+\frac{1}{2}\beta T^{2}.$$

This quadratic equation for T yields only one positive root, so that the maximum division rate for the protist is(24)$$\Omega_{1+2}=\frac{1}{T}=\frac{\Omega_{1}+\Omega_{2}}{2}\left[\sqrt{1+\frac{2\beta }{{\left(\Omega_{1}+\Omega_{2}\right)}^{2}}}+1\right].$$

Here, we use the subscript 1+2 to emphasize the synergistic nature of the estimate for $\Omega_{1+2}$. Note that $\Omega_{1+2}$ is larger than $\Omega_{1}+\Omega_{2}$, and the difference is(25)$$\Delta \Omega =\Omega_{1+2}-\left(\Omega_{1}+\Omega_{2}\right)=\frac{\Omega_{1}+\Omega_{2}}{2}\left[\sqrt{1+\frac{2\beta }{{\left(\Omega_{1}+\Omega_{2}\right)}^{2}}}-1\right].$$

### 3.2. Kleptoplastidy Index

In this subsection, we consider environments rich in both resources and focus on evaluating the relative contribution of kleptoplastidy to the division rate. By combining (24) and (25), we introduce the kleptoplastidy index *κ* defined as(26)$$\kappa :=\frac{\Delta \Omega }{\Omega_{1+2}}=\frac{\sqrt{1+\frac{2\beta }{{\left(\Omega_{1}+\Omega_{2}\right)}^{2}}}-1}{\sqrt{1+\frac{2\beta }{{\left(\Omega_{1}+\Omega_{2}\right)}^{2}}}+1}.$$

In what follows, we use this index as the measure of kleptoplastidy.

While it is clear that 0≤κ<1, one can attain a better understanding for the range of κ by introducing the dimensionless parameters(27)$$\alpha =\frac{\beta }{\Omega_{1}\Omega_{2}}=\gamma \left[\left(\frac{m_{1}}{\tau \Omega_{2}}\right)÷M_{1}\right]$$
and(28)$$\sigma =\frac{4\Omega_{1}\Omega_{2}}{{\left(\Omega_{1}+\Omega_{2}\right)}^{2}},$$
so that (26) takes the form(29)$$\kappa =\frac{\sqrt{1+\frac{1}{2}\alpha \sigma }-1}{\sqrt{1+\frac{1}{2}\alpha \sigma }+1}.$$

This form is useful for two reasons. First, it involves only dimensionless parameters. Second, both α and σ have sound physical and mathematical interpretations.

The physical meaning of α becomes clear once it is noted that in (27) the term in parentheses is the mass of kleptoplasts accumulated in the second thought experiment. Therefore, the ratio in the square brackets is the mass of kleptoplasts acquired in the second thought experiment to the mass of native chloroplasts. Since $\gamma :=\omega_{1}^{*}/\omega_{1}$, the parameter α is the ratio of the photosynthetic contributions to the division rate of kleptoplasts and native chloroplasts.

The parameter σ represents both $\Omega_{1}$ and $\Omega_{2}$. Further, it reaches its maximum, σ=1, for what one may refer to as perfectly balanced mixotrophy, characterized by $\Omega_{1}=\Omega_{2}$. In contrast, for autotrophs $\left(\Omega_{2}=0\right)$ or heterotrophs $\left(\Omega_{1}=0\right)$, the parameter attains its minimum, σ=0. Note that the dependence of σ on $\Omega_{1}$ and $\Omega_{2}$ is symmetric. This means that σ does not favor one feeding mode over the other. Rather, it represents how balanced the two feeding modes are.

In Figure 2, the kleptoplastidy index κ is plotted as a function of α and σ for 0≤α≤5 and 0≤σ≤1. Thus, the kleptoplastidy index κ is an increasing function of α and σ, and 0≤κ<1. It is clear from (29) and the plot that the maximum κ is reached when each α and σ attain their respective maximum. In particular, κ≈0.3 for α=5 and σ=1. That is, kleptoplastidy contributes about 30% to the division rate. For small α, κ is small because only a small mass of kleptoplasts is involved. For small σ, κ is small because balanced mixotrophy is the inherent basis for kleptoplastidy.

### 3.3. A Case Study

In this section, the developed model is applied to the dinoflagellate *Prorocentrum cordatum*, as the protist, and a cryptophyte microalga *Teleaulax* sp., as the prey cell. To emphasize the specificity, we refer to *Prorocentrum cordatum* as the P-cell rather than just a protist.

#### 3.3.1. Background

Due to its significance to algal blooms in general and often to HABs, the P-cell has been studied by multiple research groups [3,32,33,34,35,36]. Therefore, it is an excellent candidate for a case study in terms of both available data and ecological importance. An electron-microscope image of a P-cell is shown in Figure 3. There, we identify native chloroplasts, a food vacuole, and the nucleus. Figure 3 confirms the schematic in Figure 1, as native chloroplasts form a discontinuous belt adjacent to the cell wall interior, kleptoplasts are stored in food vacuoles or released to the cell interior when vacuoles collapse, and the nucleus exterior is available for accumulating kleptoplasts. The presence of food vacuoles implies consumption of relatively large prey cells via phagotrophy. This is consistent with kleptoplastidy, as prey cells should be large enough to contain potential kleptoplasts.

Mixotrophy of P-cells has been quantified for microalga *Teleaulax* sp. as prey cells by H.J. Jeong and co-authors [3], who estimated that phagotrophy contributes 15.5 ± 4.7% to the division rate of *P. cordatum*. Further, the equivalent spherical diameters of the P-cell and prey cell are *D* = 12.1 µm and *d* = 5.6 µm, respectively [33]. This means that the P-cell and prey cell are not significantly different in size, and therefore phagotrophy should proceed one prey cell at a time.

#### 3.3.2. The Maximum Division Rate and Kleptoplastidy Index

In this subsection, (24) and (29) are used for evaluating the maximum division rate during mixotrophy, $\Omega_{1+2}$, and the kleptoplastidy index, κ, for the P-cell and cryptophyte microalga *Teleaulax* sp. as prey cells. To this end, we rely on the values of $\Omega_{1}=0.22$ and $\Omega_{2}=0.38$ divisions per day [32], so that (28) yields(30)$$\sigma =\frac{4\Omega_{1}\Omega_{2}}{{\left(\Omega_{1}+\Omega_{2}\right)}^{2}}\approx 0.93.$$

Unfortunately, we could not find experimental data for estimating α. Nevertheless, one can estimate an upper bound on α directly from (27) by adopting three assumptions. First, we assume that the mass density of every constituent is the same and equal to the mass density of water, so that (27) can be rewritten as(31)$$\alpha =\gamma \left[\left(\frac{v_{1}}{\tau \Omega_{2}}\right)÷V_{1}\right],$$
where $v_{1}$ and $V_{1}$ are the volumes occupied by the chloroplasts of one prey cell and one P-cell, respectively. Second, the entire volume available for kleptoplasts is fully filled. Third, the kleptoplasts are as effective as native chloroplasts, so that γ=1. As a result, we obtain(32)$$\alpha_{\max}=\frac{V_{f}}{V_{1}}\approx 10,$$
where $V_{f}$ is the volume of the P-cell available for accumulating kleptoplasts, shown in green in Figure 1. The estimate $V_{f}/V_{1}\approx 10$ is obtained upon visual inspection of Figure 3.

For α=10 and σ=0.93, (24) and (29) yield(33)$$\Omega_{1+2}=\frac{\Omega_{1}+\Omega_{2}}{2}\left(\sqrt{1+\frac{\alpha \sigma }{2}}+1\right)\approx 1\;\mathrm{division}\;\mathrm{per}\;\mathrm{day}$$
and(34)κ≈0.4.

These two estimates imply that kleptoplastidy can significantly accelerate reproductive growth of mixotrophic protists.

It is instructive to plot the history of concentration for various conditions. To this end, we assume exponential growth. In Figure 4, we plot the current concentration c normalized by the initial concentration $c_{0}$ as a function of time for a one-week period; note that the log scale is used along the ordinate. We consider four situations:(1)The environment is inorganic-resource-rich, but it contains no organic resources. In this case, the division rate is $\Omega_{1}=0.22$ divisions per day.(2)The environment is organic-resource-rich, but it contains no inorganic resources. In this case, the division rate is $\Omega_{2}=0.38$ divisions per day.(3)The environment is rich in both resources, but kleptoplastidy is suppressed. In this case, the division rate is $\Omega_{1}+\Omega_{2}=0.60$ divisions per day.(4)The environment is rich in both resources, and kleptoplastidy is operational. In this case, the division rate is $\Omega_{1+2}=1$ division per day.

Thus, Figure 4 clearly demonstrates that kleptoplastidy is a potent mechanism, capable of increasing the cells concentration by three orders of magnitude in environments rich in both resources. In contrast, in environments rich only in one resource, where kleptoplastidy is suppressed, the gain is only one order of magnitude.

## 4. Discussion

### 4.1. Interpretation of the Main Findings

In this paper, guided by the topical issue of unveiling cellular and molecular drivers of algal blooms’ formation, we developed a simple mathematical model for mixotrophic protists capable of kleptoplastidy. Since kleptoplastidy has been observed in many aquatic species ranging from microalgae to slugs [39,40,41,42,43], we believe that the model or its variants can be applicable to other aquatic species.

The model was developed using a generalization of Monod’s equations and three thought experiments involving controlled environments, which can be realized in laboratory settings only. The first (second) experiment was designed so that only the autotrophic (heterotrophic) mode was active, and in the third experiment all three modes, the autotrophic, heterotrophic, and kleptoplastidic, were active. As a result, in processing experimental results, it was possible to isolate the kleptoplastidic mode; see (17), (19), and (20). The environments were either resource-rich (*r*_1_, *r*_2_ >> 1) or resource-poor $\left(r_{1},r_{2}=0\right)$, which was sufficient for our purposes because we were focused on the maximum division rates, believed to be associated with HABs. Using controlled environments with moderate resource concentrations $\left(r_{1},r_{2}=O\left(1\right)\right)$ makes modeling more complicated as (5) is no longer reduced to (12) or (14), but rather to equations involving $r_{1}$ and $r_{2}$. However, environments with moderate resource concentrations are also beneficial, as they allow one to determine the half-saturation constants $K_{1}$ and $K_{2}$ from the first and second thought experiments. Then, the third experiment can be modeled by generalizing (17) as(35)$$\mu^{\left(3\right)}=\Omega_{1}\frac{R_{1}}{K_{1}+R_{1}}+\Omega_{2}\frac{R_{2}}{K_{2}+R_{2}}+\beta t\frac{R_{1}}{K_{1}+R_{1}}\frac{R_{2}}{K_{2}+R_{2}}.$$

The validity of this equation is impossible to establish without adequate experimental data. But the need for such an equation is apparent, especially for improving existing mixotrophy models and HABs forecasting [24,25,26,27,28,29,30,44].

The model was developed for one protist, one inorganic resource, and one organic resource. Formally, it is straightforward to extend to N protist species, $N_{1}$ inorganic resource types, and $N_{2}$ organic resource types. To this end, we rewrite (17) as(36)$$\mu_{i}=\sum_{i_{1}=1}^{N_{1}}\Omega_{1i_{1}}+\sum_{i_{2}=1}^{N_{2}}\Omega_{2i_{2}}+\sum_{i_{2}=1}^{N_{2}}\sum_{i_{1}=1}^{N_{1}}\beta_{i_{1}i_{2}}t\;\;\;\;\;\;i=1,\ldots ,N.$$

For each i, this equation contains $N_{1}$ constants in the first sum for modeling autotrophy, $N_{2}$ constants in the second sum for modeling heterotrophy, and $N_{1}N_{2}$ constants in the double sum for modeling kleptoplastidy. Thus, in total, the model involves $N\left(N_{1}+N_{2}+N_{1}N_{2}\right)$ parameters and requires the same number of experiments, all of which are structured like the thought experiments in Section 3.1.4. If desired, one can combine (35) and (36) to derive a generalization of (36) valid for moderate resource concentrations. In this case, the number of parameters and required experiments increases to $N\left(2N_{1}+2N_{2}+N_{1}N_{2}\right)$.

In the second and third experiment, it was assumed that kleptoplastidy is not affected by saturation of the free volume by kleptoplasts. This assumption needs to be questioned, especially considering possibilities of protistan daughter cells inheriting kleptoplasts of their parent cells. If the saturation occurs, one ends up with interesting possibilities and much more significant modeling challenges. For example, a saturated protist may behave as an autotroph, as it has no room for accommodating prey cells. Alternatively, a saturated protist may perform as a mixotroph, but instead of relying on phagotrophy, it switches to osmotrophy and consumption of dissolved organic materials (DOM). Mathematically, such switches in the feeding strategy are associated with nonlinearities. In this regard, the proposed model is applicable to situations occurring well before the saturation point is reached.

The estimate for α, adopted for calculating (33) and (34), should be regarded as an upper bound, as it assumes that (i) the free space is saturated with kleptoplasts and (ii) γ=1. The latter assumption is particularly optimistic, as native chloroplasts are located near the surface, in positions advantageous for photosynthesis (Figure 1 and Figure 3). Thus, kleptoplasts are relegated to positions away from the surface, with inferior conditions for photosynthesis, and therefore one should expect γ<1.

The kleptoplastidy index κ is rather insensitive to small variations in the input parameters α and σ. One way of establishing this is by showing that the absolute values of the first derivatives of κ with respect to these parameters are sufficiently small [45]. In the current setting, this requirement translates into the inequalities $\left|\partial \kappa /\partial \alpha \right|<1$ and $\left|\partial \kappa /\partial \sigma \right|<1$. For simplicity, we show this for σ=1, as this restriction is the most important for kleptoplastidy. Upon differentiation of κ, we obtain(37)$${\frac{\partial \kappa }{\partial \alpha }}_{|\sigma =1}=\frac{1}{2\sqrt{1+\frac{\alpha }{2}}{\left(\sqrt{1+\frac{\alpha }{2}}+1\right)}^{2}}\;\mathrm{and}\;{\frac{\partial \kappa }{\partial \sigma }}_{|\sigma =1}=\frac{\alpha }{2\sqrt{1+\frac{\alpha }{2}}{\left(\sqrt{1+\frac{\alpha }{2}}+1\right)}^{2}}.$$

The bounds for each derivative can be found by finding its minimum and maximum. Accordingly, using basic calculus, we determine that $0\leq \partial \kappa /\partial \alpha_{|\sigma =1}\leq 1/8$ and $0\leq {\partial \kappa /\partial \sigma }_{|\sigma =1}<0.172$. Thus, the absolute values of both derivatives are less than one, and therefore the model is stable.

The estimates in Section 3.3 are limited to environments rich in both inorganic and organic resources. While their generalization to environments with moderate resource concentrations needs to be supported experimentally, we can evaluate situations when the environment changes from being rich only in one resource to being rich in both resources. In particular, if the environment changes from being inorganic-resource-rich to being rich in both resources, then the division rate switches from $\Omega_{1}=0.22$ to $\Omega_{1+2}=1$ divisions per day. This is more than 4.5 times increase. Similarly, if the environment changes from being organic-resource-rich to being rich in both resources, then the division rate switches from $\Omega_{2}=0.38$ to $\Omega_{1+2}=1$ divisions per day. This is more than 2.5 times increase. Both increases are significant, especially if the reproductive growth process is exponential. These increases in the cells’ division rate may result in dramatic mixoplankton population growth, as shown in Figure 4.

### 4.2. Advantages and Limitations of the Study

The results of this study contribute to the theoretical biology by mathematical modeling of kleptoplastidy, regarded as a key component of the mixotrophic feeding strategy in protists [39,40,41,42,43,44]. This reveals another aspect explaining high adaptability of such organisms to fluctuating environments [1,3,4,5,6], due to wider trophic niches of mixoplankton compared to photoautotrophic plankton [4,5,8,9,10,34,35,36,44]. In addition, this allows one to regard kleptoplastidy as a harmful bloom forming mechanism [30]. To the best of our knowledge, the proposed model is the only one, which clearly demonstrates how kleptoplastidy accelerates the growth of mixotrophic algae populations. The model focuses on two quantities of interest. First, the division rate $\Omega_{1+2}$ of bloom-forming mixotrophic algae capable of kleptoplastidy. This quantity is essential for evaluating and forecasting algal blooms in general, and HABs in particular. Second, the kleptoplastidy index κ, or the relative contribution of kleptoplastidy to $\Omega_{1+2}$, which allows one to assess how strongly bloom-forming mixotrophic algae react to changes in nutrients’ composition and abundance in the environment. Therefore, the first numerical assessment of the kleptoplastidy contribution to population growth of bloom-forming mixotrophic protists by calculating the kleptoplastidy index is a new and advantageous approach to comprehension, evaluation and forecasting algal blooms in general, and HABs in particular.

The limitations of the proposed model are due to scarcity of experimental data on growth rates of bloom-forming mixotrophic protists in environments suitable for mixotrophy [44]. For this reason, our case study (Section 3.3) was carried out using only one bloom-forming potentially toxic dinoflagellate species, *Prorocentrum cordatum*, as the protist, and the cryptophyte microalga *Teleaulax* sp., as the prey cell. These calculations are based on several literature sources [3,32,33] that, to our knowledge, contain the most complete and representative experimental data on the physiological characteristics and mixotrophic feeding of *P. cordatum*, our major species of interest [8,20,21,34,35,36,44]. Therefore, we believe that usage of these particular sources of data secured the most adequate results of our case study. Consequently, the validation of our analytical model is the task for future experimental investigations.

### 4.3. Future Perspectives

The issue of the productive capabilities of organisms is one of the key questions in modern ecology [2,3,4,5,6,11,12]. A number of different disciplines in natural sciences require information on fine mechanisms that drive life in aquatic ecosystems [44]. Specifically, the processes that ensure high feeding efficiency are among basic prerequisites of successful performance of organisms, their populations and communities. This can be proven by a large number of publications demonstrating the enhancement and worldwide spreading of HABs in marine coastal ecosystems and their dependence on trophic and physical-chemical conditions varying in space and time [2,10,11,12,13,14]. Algal blooms may result in significant shifts in community structure and functions, and as such changes can occur without a single dominant ecosystem driver, emergent ecosystem-level responses are difficult to predict [22,30]. Generally, there is a fair understanding of how different components interrelate in aquatic ecosystems in terms of who-eats-whom [46,47]. Meanwhile, the inter- and intracellular interactions within microbial communities largely remain enigmatic, as well as the question of how exactly some feeding strategies are implemented.

Our results offer a new perspective on the study of microbial mixotrophy as a HAB-forming mechanism [30] and a factor potentially influencing the productivity of phytoplankton and entire aquatic communities. In this study, kleptoplastidy as a primary though poorly studied aspect of mixotrophy was for the first time modeled, aiming at evaluation of its effectiveness. This aspect so far has never been taken into account in mathematical models of mixotrophic population growth and dynamics [19,24,26,28,44,47]. The proposed model and the new kleptoplastidy index may provide quantitative assessment of the functional role of mixotrophic protists in aquatic ecosystems. As a broader implication, the future incorporation of the kleptoplastidy model into larger ecosystem models will improve forecasts of algal blooms, including HABs, and contribute to the development of new monitoring strategies and effective environmental management of marine coastal ecosystems.

## 5. Conclusions

In this study, a basic mathematical model of kleptoplastidy was first developed. The model demonstrates how kleptoplastidy may significantly accelerate the reproductive growth of bloom-forming mixotrophic protists and thus lead to algal blooms and, in the cases of toxic or potentially toxic protists, to HABs. The model showed that kleptoplastidy in mixotrophic protists is a synergistic process, requiring both inorganic and organic resources. Further, the model is developed for resource-rich environments, under the provision that such environments are necessary for HABs.

The new model gives rise to the kleptoplastidy index *κ*, a new ecological indicator, which allows one to assess how kleptoplastidy contributes to proliferation of mixotrophic protists. This indicator is a function of the parameters *α* and *σ*. The former represents the photosynthetic contribution of kleptoplasts normalized by that of native chloroplasts. The latter represents the balance between the autotrophic and heterotrophic feeding modes.

The kleptoplastidy phenomenon and possibility of its ecological indication in terms of the kleptoplastidy index *κ* was illustrated by using existing experimental data for potentially toxic bloom-forming mixotrophic dinoflagellates *Prorocentrum cordatum* feeding on the cryptophyte microalga *Teleaulax* sp. For this system, we calculated and compared the cells’ division rates in various resource-rich environments. Our modeling results support the hypothesis that kleptoplastidy, as a feeding mode, can increase the division rate of mixotrophic protists significantly (by up to 40%), thus boosting their population growth and, consequently, promoting algal blooms.

The new developments can contribute to advancements in mathematical modeling of cellular mechanisms that underpin basic ecological processes of utmost importance—the formation of HABs that deteriorate marine and coastal environments worldwide. Incorporation of the kleptoplastidy model into comprehensive ecosystem models will improve HABs’ forecasts and future monitoring programs thus securing effective management of aquatic ecosystems.

## Figures and Tables

**Figure 1 biology-14-00900-f001:**
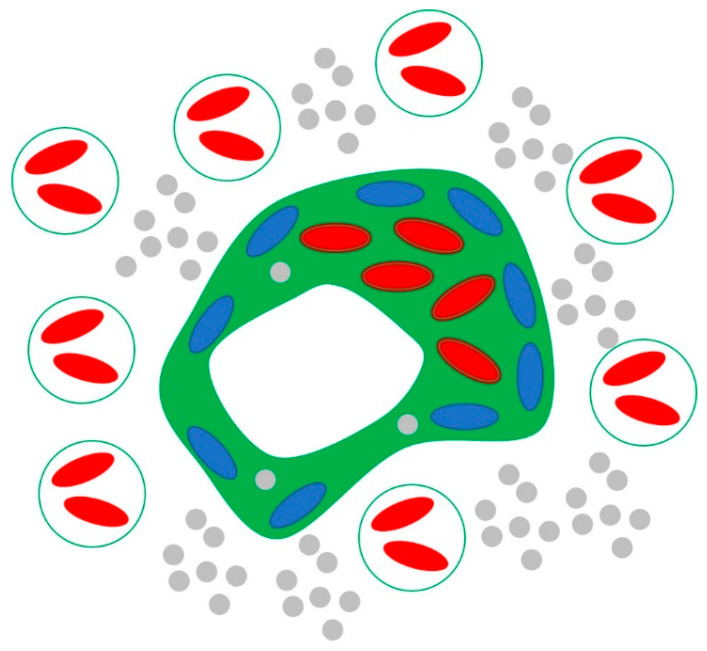
The conventional reference mixotrophic protist (large irregular shape), inorganic resources (small grey circles), and prey cells (large circles). Native chloroplasts of the protist are shown in blue, native chloroplasts of prey cells are shown in red, and kleptoplasts are shown in red with dark outline. The volume available for acquiring kleptoplasts is shown in green.

**Figure 2 biology-14-00900-f002:**
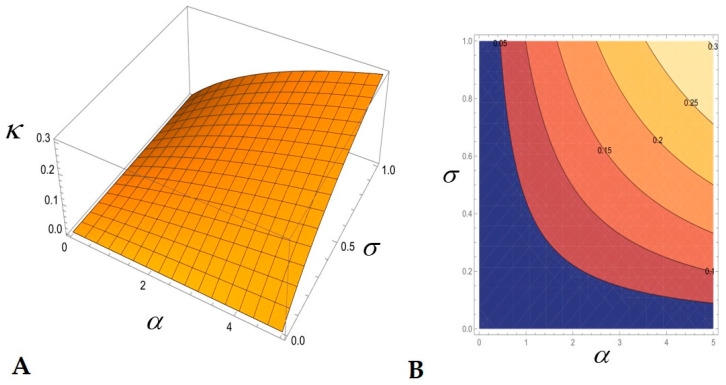
The kleptoplastidy index κ as a function of the dimensionless parameters α and σ: (**A**) 3-D plot, (**B**) contour plot (color gradient and numbers at the lines indicate the κ values: <0.05 in blue color, >0.25 in light yellow). The parameter α represents the photosynthetic contribution of kleptoplasts normalized by that of native chloroplasts. The parameter σ represents the balance between the autotrophic and heterotrophic feeding modes.

**Figure 3 biology-14-00900-f003:**
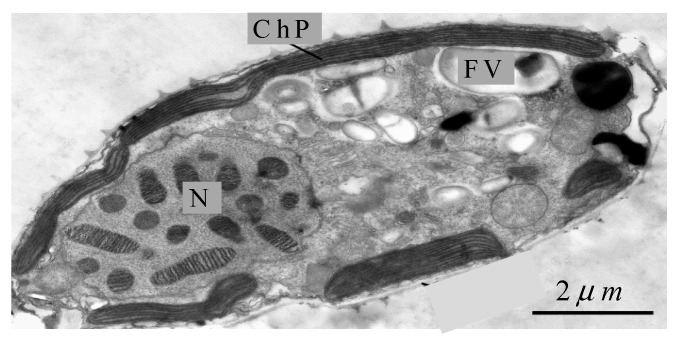
An electron-microscopy image of *Prorocentrum cordatum* (photo courtesy of M.A. Berdieva). Abbreviations: FV—food vacuole, N—nucleus, ChP—native chloroplasts of the P-cell.

**Figure 4 biology-14-00900-f004:**
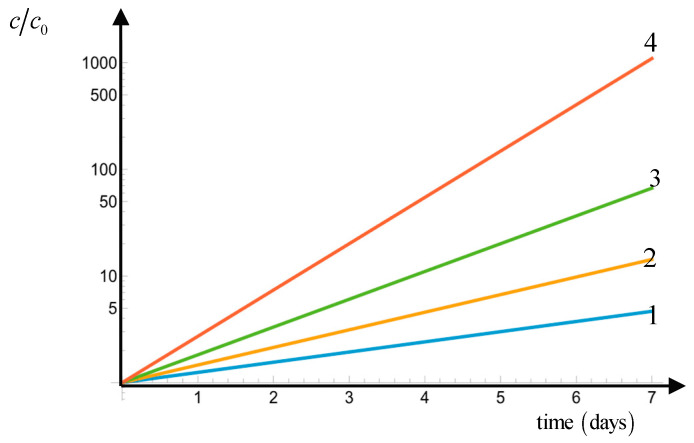
The normalized concentration of protist cells $c/c_{0}$ (log scale) as a function of time: (1) the environment is inorganic-resource-rich, but it contains no organic resources, (2) the environment is organic-resource-rich, but it contains no inorganic resources, (3) the environment is rich in both resources, but kleptoplastidy is suppressed, and (4) the environment is rich in both resources, and kleptoplastidy is operational.

**Table 1 biology-14-00900-t001:** Nomenclature.

Symbol	Description	Units
μ	A generic division rate of the protist	day^−1^
$\mu^{\left(1\right)}$	The protist division rate measured in the first thought experiment	day^−1^
$\mu^{\left(2\right)}$	The protist division rate measured in the second thought experiment	day^−1^
$\mu^{\left(3\right)}$	The protist division rate measured in the third thought experiment	day^−1^
Ω	A generic maximum division rate of the protist	day^−1^
$\Omega_{1}$	The maximum protist division rate due to inorganic resources	day^−1^
$\Omega_{2}$	The maximum protist division rate due to organic resources	day^−1^
$\Omega_{1+2}$	The maximum protist division rate due to inorganic and organic resources in the presence of kleptoplastidy	day^−1^
R	The concentration of a generic resource	µg L^–1^
$R_{1}$	The concentration of an inorganic resource	µg L^–1^
$R_{2}$	The concentration of an organic resource	µg L^–1^
K	A generic half-saturation constant	µg L^–1^
$K_{1}$	The half-saturation constant for inorganic resources	µg L^–1^
$K_{2}$	The half-saturation constant for organic resources	µg L^–1^
r	The normalized concentration of a generic resource	none
$r_{1}$	The normalized concentration of an inorganic resource	none
$r_{2}$	The normalized concentration of an organic resource	none
ψ	Biological time	none
*t*	Time	days
*T*	The time interval between two divisions of the protist	days
*τ*	The time interval for digesting one prey cell by the protist	days
$m_{1}$	The mass of chloroplasts in one prey cell	µg
$M_{1}$	The mass of native chloroplasts in the protist	µg
$\omega_{1}$	The maximum division rate of the protist due to a unit native chloroplast mass	µg^–1^ day^–1^
$\omega_{1}^{*}$	The maximum division rate of the protist due to a unit kleptoplast mass	µg^–1^ day^–1^
γ	The maximum division rate of the protist due to a unit kleptoplast mass normalized by that of the native chloroplasts	none
β	The phenomenological parameter representing the acceleration of the maximum division rate due to kleptoplastidy	µg^–1^ day^–2^
*k*	Kleptoplastidy index	none
*α*	The photosynthetic contribution of kleptoplasts normalized by that of native chloroplasts	none
*σ*	The balance parameter between the autotrophic and heterotrophic feeding modes of the reference protist	none
$v_{1}$	The volume of the chloroplasts of one prey cell	µm^3^
$V_{1}$	The volume of the chloroplasts of the protist	µm^3^

## Data Availability

The study does not report any data.

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
