# Peer review of "Modeling Unveils How Kleptoplastidy Affects Mixotrophy Boosting Algal Blooms"

_biology, 2025, doi:10.3390/biology14070900_

Round 1

Reviewer 1 Report

Comments and Suggestions for Authors

The issue of the productive capabilities of organisms is one of the key questions in modern ecology. This manuscript offers a new perspective on the study of microbial mixotrophy as a factor potentially influencing the productivity of entire communities.

The manuscript appears to be a complete and coherent study. In my opinion, the following suggestions could strengthen the connection between the theoretical propositions and ecological research on microorganisms in aquatic ecosystems:

  1. It would be helpful to include a separate subsection in the model description that summarizes the limitations of the proposed model.
  2. In Section 3.6.2, the calculation of the kleptoplasty index is based on data from a single literature source. Are the data necessary for this calculation available only in that one publication? It appears that the authors used just one study to perform the calculation and concluded that kleptoplasty can significantly accelerate reproductive growth of mixotrophic protists. It would be more robust to include at least a couple more examples for the calculation, assuming such data are available in the global literature.
  3. The discussion section largely repeats the introduction and the results. It would also be useful to include a discussion on how the authors’ findings could impact the study of algal blooms in specific water bodies—for instance, how the ratio of organic to inorganic nutrients might influence the scale of blooms, particularly depending on whether the bloom-forming organism is capable of mixotrophy.

Reviewer 2 Report

Comments and Suggestions for Authors

General comments

The manuscript focuses on mathematical modeling of kleptoplastidy. This is a sophisticated and largely underestimated cellular-molecular process. However, this phenomenon is vitally important for supporting algal growth rates and can significantly enhance their proliferation (by up to 40%, as shown in this study). The numerical model developed by the authors allowed proving their hypothesis, which suggests that in the cases of bloom-forming algae, kleptoplastidy can contribute substantially to the formation of algal blooms; meanwhile, the latter can often be harmful for the environment. Of particular importance is the fact that the authors illustrated their model using the data on the globally distributed, common species of potentially toxic, bloom forming dinoflagellates. Studies of such blooms is the hot issue of microbial ecology nowadays, as these algal outbreaks are often toxic and deteriorate aquatic communities. Meanwhile, modeling, prediction and management of algal blooms are very difficult tasks, mainly because many fine-scale processes that underpin blooms, such as kleptoplastidy, are still poorly understood and their evaluation is lacking. In this context, the current research is of special value. In particular, I admit that the developed new ecological indicator, the kleptoplastidy index, will be widely used in the future mathematical modeling of basic ecosystem processes, including harmful algal blooms.

I also highly appreciate the strong theoretical part of this research. The authors in many details consider some basic nutritive pathways of the chloroplast-containing protist cells in three well developed “thought experiments”. These theoretical experiments, coupled with the concept of biological time, allowed the authors to overcome the problem of insufficiency of natural experimental data on kleptoplastidy in their model organisms.

I can suggest only some minor improvements of the presentation of the material in this manuscript. My specific comments and suggestions for the authors are listed below.

Specific comments

1) The manuscript has the classical structure, which is always difficult to achieve when presenting a mathematical model. Meanwhile, on the one hand, I think that the information in the Materials and Methods is somewhat too general, and more methodological details could be included in this section. However, on the other hand, it is very helpful for reading and comprehending the model if its background and major methodological details are provided directly next to the formulas and equations, as it was originally done by the authors. Therefore, I do not insist on these amendments.

2) The text in lines 130-135 repeats the content of the legend to Figure 1. I suggest deleting this duplication or at least shortening this section in the text, keeping all the details in the figure legend.

3) In the legend to Figure 2 (line 383), I suggest to add brief explanations of the parameters alpha and sigma, to make this figure self-explainable, as it contains one of the major results of the study.

4) In the Discussion, I would like to see more information about the novelty of the authors’ approach in comparison with the available models of mixotrophy in protists. Some of these models are cited in the article (e.g., references ## 24-26, 28, 29, etc.), but they seem to be not discussed in sufficient details.

5) It would be good to indicate briefly the main limitations of the study; this could be a separate paragraph in the Discussion.

Reviewer 3 Report

Comments and Suggestions for Authors

Title

Consider changing the title. 

Consider changing the title. Maybe one of these?

  1. The Role of Kleptoplastidy in Mixotrophic Growth"
  2. Kleptoplastidy’s Impact on Mixotrophy and Algal Success"
  3. Modeling Kleptoplastidy’s Role in Mixotrophic Algae"
  4. How Kleptoplastidy Drives Mixotrophic Algal Dominance"

Simple Summary

Rewrite SS section

SS contains too many specific terms like @ photosynthetic reactors”, phagotrophic engulfment, ecological indicator  etc. Please, simplify the SS section, make it readable for border auditory, not only specialists. Now it looks like an abstract without simplification of technical language. 

SS is too long and dence now, long and compound sentences. SS includes lots of details that are suitable for abstract or main text. The relevance for the real world is unclear now. Start and conclude SS with the impact of your study for ecosystems. 

Abstract

rewrite Abstract section. It's too long and dense now. Break long sentences into several shorter ones. Please, reorganize logical flow: Intro-hypothesis-methods-results-conclusions/significance.

The sentence “constructed using three thought experiments and the concept of biological time” (line 38) is difficult to understand, rewrite it.

The phrase “Can significantly increase (by up to 40%) the division rate of algae” (line 44) - is also unclear. Was this modeled or validated with empirical data?

Simplify phrases:

“ratio of photosynthetic production of acquired versus native chloroplasts” 

 “balance between autotrophy and heterotrophy” 

They are not reader-friendly]

Please, define or explain the terms used in phrases “a simple mathematical model” , “thought experiments”.

The abstract now doesn’t clearly say what is new  and what was improved during your research. 

Conclude abstract with sentence showing impact of your research and the benefits for harmful algal blooms prediction. 

key words

fine

Introduction

Rewrite the introduction

It provides an extensive overview of mixotrophy and kleptoplastidity, but doesnt say what is missing in this part of knowledge. 

Line 52 “kleptoplastidy is poorly studied” and how  this lack impacts the research field.

Please, state the hypothesis of research aim earlier, not at the end of Intro. Now its difficult to understand the research goal.

The first three paragraphs line 52-71 are overloaded with information (lots of definitions, types of kleptoplastidity, citations). Please, rewrite focusing on why kleptoplastidity is understudied and ecologically significant.

Overall, Introduction needs better focusing, prioritisation of information and demonstration of  research gap and novelty. 

Materials and Methods

Include a clear description of  experimental datasets, experimental conditions, replicates, source articles and others..

Specify where experimental data were obtained, which data were actually used, how data were curated or validated.

Now there are insufficient details for reproducibility. Please add subsections with details for  model assumptions, key equations, parameter definitions, data input values..

Math framework described only conceptually , no equations, parameter values, code references are given.

Line 94 “thought experiments lack definitions and parameter assignments. 

Clarify are thought experiments based on literature values or actual experiments? Were there any empirical validation?

Lines 95-99 refer to three hypothetical experiments. However no equations or conditions were defined,

Were these experiments real or theoretical?

Include a table or list with biological parameters, including symbol, biological meaning, units, source or reference.

Line 92 refers to “measurable, at least in principle, parameters” , but does not specify them. No units, expected ranges or justification are present.

What computational model did you use? Mention it. How this model was implemented: programming language, any simulations, any code is available?

What your model predicts and simulates and what was the aim of this particular model?

Results

Please, separate model development and results. Please, move model derivation into a separate section model”

Seek and incorporate empirical data for model validation.

Results are poorly labeled with no clear plots.

Add growth curves for P. cordatum with and without kleptoplastidity

Add plot showing how k changes across α and σ .

Results are mathematically dense with no biological implication. After each formula briefly explain its biological meaning. 

Please, expand case study with a summarising table (inputs and outputs); explore how k is sensitive to chloroplast density Ë Ñ‚

Add biological discussion

Add population -level extrapolation

State the limitations clearly

Link to harmful algal blooms, you address it in abstract and intro sections and no info in the results or discussion.

Figures

Are mostly conceptual, but not data driven. Figures don't show simulation results or model output. Figures are not referenced or discussed clearly. 

Figure_1

add clear labels

add scale bar

Cite and interpret in the text.

Figure_2 

represents a surface plot of kleptoplastidity index k as a function

add labels for axis units

Add color-scale legend

Interpret the plot deeply in text

Explain significance of the plot for non-mathematician reader.

Figure_3

Modified from a prior source but exact credit and permission are unclear. Add permission status. 

This image does not clearly align with figure 1.

For all figures:

add complete and self-sufficient legends and explanations. 

Explain what each figure shows, how it supports the model and why it is important.

Add side-by-side microscopic and schematic comparisons to link morphology and theory.

Discussion

Rewrite discussion section. At least, reorganize it for logical flow. 

  1. Interpret main findings of research
  2. Compare your data with literature 
  3. discuss model limitations
  4. speculate and discuss broader implications
  5. Conclude with future perspectives

The discussion section is now overfocused on reexplaining the model. Much of the discussion repeats the description of the model instead of explaining the biological meaning of results. There is little distinction between what was done and its biological meaning. 

Add a paragraph comparing this model to previous ones, citing reviews, Right now there is no critical comparison between your model and previous HAb and mixotrophy models.

Add an explanation of how kleptoplastidity might impact on  nutrient cycles, food web interaction,  adaptation to changing environments. 

Add analysis of how sensitive the model is to assumptions about α and γ. Are there any possible errors or uncertainty introduced by them?

Add practical implications for monitoring and management

Conclusion

rewrite conclusion section. Now it repeats the information from results snd abstract. Use conclusions to synthesize the main points and future perspectives for the research field. Add practical implications. Add what is new in your study. Introduction of the first model of kleptoplastidity or first kleptoplastidity index?

Literature cited

Double-check references for consistency using Crossref tool

Reviewer 4 Report

Comments and Suggestions for Authors

The paper represents a valuable contribution to the theoretical understanding and mathematical modeling of mixotrophy and kleptoplastidy in protists. The proposed model has the merit of suggesting mechanisms through which kleptoplastidy can accelerate the growth of mixotrophic populations, and implicitly, provides an additional mechanistic explanation for the formation of algal blooms. The study aims to apply this model to real data (in the case of the dinoflagellate Prorocentrum cordatum) to demonstrate its validity and usefulness.

The theoretical development of an ecological indicator is an important step, and the use of accumulated and published data to develop correlations and hypotheses is commendable and should be encouraged.
I believe that a numerical simulation would allow the authors to demonstrate the behavior of the indicator, illustrate its applicability, and highlight any potential limitations or advantages (or, in the future, to determine the applicability limits of the developed indicator).

If a simulation is not feasible, I consider it useful to test the proposed indicator (κ) with other mathematical models as well, to demonstrate that the results do not strictly depend on the choice of the initial model. This approach would strengthen the robustness of the conclusions and the scientific credibility of the indicator.
Of course, the condition of simultaneous abundance of organic and inorganic resources should be maintained. But I do not insist.

Round 2

Reviewer 3 Report

Comments and Suggestions for Authors

Simple summary

The SS section shows notable improvements. The language was simplified, sentences became shorter. However, I do have several comments. Start the SS section with sentences engaging about why algal blooms matter globally. Briefly explain why this model is new?

Abstract

Improved. Now sound better.

Introduction

Notable improvements

Materials and methods

There are several improvements. However, I have several comments:

The section still lacks parameter table (symbols, units, data sources for parameters, ranges)

Without this  and without a summary of key equations it is still difficult for the reader to reproduce the result. 

Results

The section was drastically improved! However it is still difficult for non-matematitian readers.

It would benefit from adding additional 2D projections with color gradients. 

Discussion

This section obviously was improved. The authors made a great job.  However, the discussion would benefit from focusing on ecological application rather than  model mechanics; adding real examples of managing HABs and suggesting next steps for experimental validation.

Conclusions

Add examples of how practitioners could use model or indexes

Overall, I think authors made a huge job and manuscript definetely should be accepted at present or minor revisions. I do understand that polishing paper might be an endless process.
